# Test-Time Mixup Augmentation for
# Uncertainty Estimation in Skin Lesion Diagnosis

**Hansang Lee**[1]                                    HANSANGLEE@KAIST.AC.KR

**Haeil Lee**[1]                                      HAEIL.LEE@KAIST.AC.KR

**Helen Hong**[*2]                                    HLHONG@SWU.AC.KR

**Junmo Kim**[1]                                      JUNMO.KIM@KAIST.AC.KR

[1] *School of Electrical Engineering, Korea Advanced Institute of Science and Technology, Republic of Korea*

[2] *Department of Software Convergence, Seoul Women's University, Republic of Korea*

**Editors:** Under Review for MIDL 2021

## Abstract

Uncertainty is considered to be an important measure that provides valuable information on the learning behavior of deep neural networks. In this paper, we propose an uncertainty estimation method using test-time mixup augmentation (TTMA). The TTMA uncertainty is obtained by replacing affine augmentation with the mixup in the existing test-time augmentation (TTA) method. In addition to the data uncertainty, we propose TTMA-based class-specific uncertainty, which can provide information on between-class confusion. In experiments on the skin lesion diagnosis dataset, we confirmed that the proposed TTMA not only provides better epistemic uncertainty than TTA but also provides information on between-class confusion through class-specific uncertainty.

**Keywords:** Uncertainty estimation, mixup, data augmentation, skin lesion diagnosis.

## 1. Introduction

Uncertainty estimation, which measures how much confidence a deep learning model has about its decisions, has received attention in recent years. Uncertainty is not only used to improve the algorithm efficiency through feedback but also provides information on the reliability of the learning models, such as being given to clinicians in the medical applications. Test-time-augmentation (TTA) (Wang et al., 2019) is a method of uncertainty estimation by giving perturbation on the test data through affine augmentation and measuring the entropy of prediction results. TTA is widely used as an uncertainty estimation along with MC Dropout due to its ease of implementation. However, while TTA uncertainty is sensitive in *aleatoric uncertainty* in response to the data perturbation, it is less sensitive in *epistemic uncertainty* in response to the out-of-distribution (OoD) data.

In this paper, we propose a method of uncertainty estimation with test-time mixup augmentation (TTMA) by replacing affine augmentation with mixup method (Zhang et al., 2017) in the existing TTA. Mixup plays a role in regularizing the learner for class boundary regions by mixing the data from different classes. Using these characteristics of mixup, the proposed TTMA can provide not only better epistemic uncertainty, which is sensitive to OoD data than TTA, but also *class-specific uncertainty* to determine the between-class confusion.

---

[*] Corresponding author

## 2. Methods

The proposed method consists of two elements: (1) data uncertainty with TTMA and (2) class-specific uncertainty. In data uncertainty estimation with TTMA, we apply mixup augmentation on test data to obtain the perturbation-robust results and estimate the uncertainty. For a given test data $x$, we form a mixed test data $\tilde{x}_{mj} = \alpha x + (1 - \alpha)x_{mj}$ by mixing $x$ with randomly selected training data $x_{mj}$, where $x_{mj}$ is a $j$-th randomly selected data from the training set of class $m = 1, ..., M$. According to the assumption of mixup, the soft label of the mixed data $\tilde{x}_{mj}$ can also be expressed as a linear combination of two labels, i.e., $\tilde{y}_{mj} = \alpha y + (1-\alpha)y_{mj}$, where $y$ and $y_{mj}$ are the ground truth labels of $x$ and $x_{mj}$, respectively. Using this formula, we can infer the label $\hat{y}$ of the original test data $x$ associated with $x_{mj}$ from the prediction result of mixed data $f(\tilde{x}_{mj})$ as $\hat{y}|_{mj} = (f(\tilde{x}_{mj}) - (1-\alpha)y_{mj})/\alpha$. The final test label $\hat{y}$ can be obtained by majority voting from $\{\hat{y}|_{mj}|m = 1, ..., M, j = 1, ..., J\}$. The data uncertainty is then computed by the entropy of the distribution of inferred labels $\{\hat{y}|_{mj}\}$. For the label frequency $p_l(\hat{y})$ whose label $\hat{y}|_{mj}$ is classified as class $l$ among $M \times J$ inferred labels, the entropy-based uncertainty is obtained by $H(y) = -\sum_{l=1}^{M} p_l(\hat{y})ln(p_l(\hat{y}))$.

In class-specific uncertainty estimation, we can define the uncertainty of test data for a specific mixup class $k$ by computing entropy for class-specific inference results. For the label frequency $p_l(\hat{y}|k)$ whose label $\hat{y}|_{kj}$ is classified as class $l$ among $J$ inferred labels, the class-specific uncertainty can be obtained by $H(y|k) = -\sum_{l=1}^{M} p_l(\hat{y}|k)ln(p_l(\hat{y}|k))$. This class-specific uncertainty can provide information of between-class confusion: If $H(y|k)$ for class $q$ is highly distributed for the data of class $p$, it can be interpreted that the distance between the class $p$ and class $q$ in the feature space is close so it is vulnerable to the perturbation between the two classes, which can be determined as "confusing."

## 3. Results and Conclusion

The proposed method was validated on the ISIC 2018 skin lesion diagnosis dataset (Codella et al., 2019). The dataset consists of 10,015 training and 193 validation images with 7 classes, and 50 squamous cell carcinoma (SCC) images from ISIC 2019 dataset were added to the validation set for OoD validation. ResNet-18 was used as the learning model. As shown in Figure 1 (a), the proposed TTMA improved both lesion diagnosis accuracy and expected calibration error (ECE) compared to TTA.

To verify the epistemic uncertainty performance of the proposed TTMA, the uncertainty distribution for the unseen class (SCC), which was not seen in the training phase, was measured and compared. As shown in Figure 1 (b) and (c), the uncertainty of SCC in the proposed TTMA is significantly higher than those of other seen classes, while the uncertainty of SCC in TTA is distributed similarly to those of seen classes. It can be confirmed that the proposed TTMA is more sensitive to OoD data than TTA and has better epistemic uncertainty behavior.

To verify the performance of the class-specific uncertainty, we compared the distribution of the class-specific uncertainty with the images of lesion cases and observed whether there is a correlation between class-specific uncertainty and between-class similarity. As shown in the boxplot in Figure 2, AKIEC and DF show higher uncertainty while NV and MEL show relatively lower uncertainty specific to BKL. This behavior of class-specific uncertainty

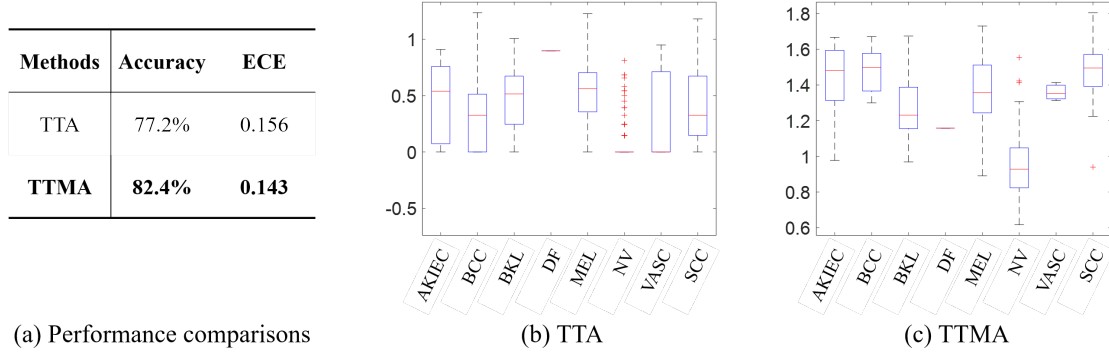

| Methods | Accuracy | ECE |
|---------|----------|-----|
| TTA | 77.2% | 0.156 |
| **TTMA** | **82.4%** | **0.143** |

(a) Performance comparisons      (b) TTA      (c) TTMA

Figure 1: Performance table (a) and the boxplots of uncertainty values (b,c).

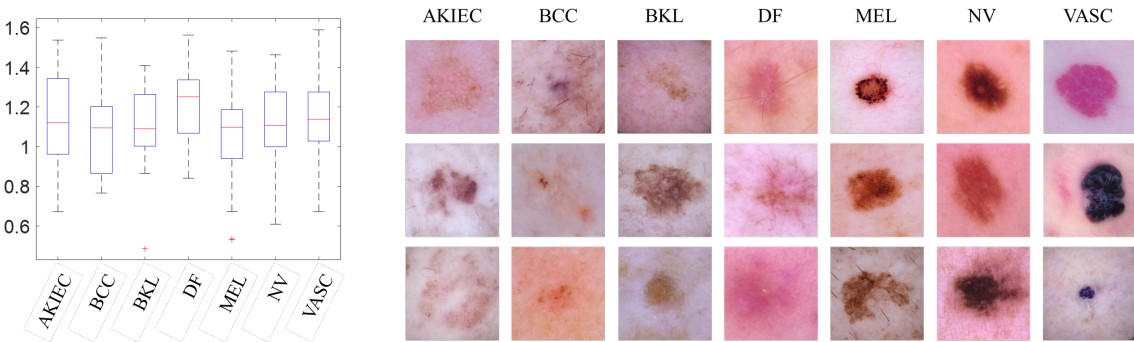

Figure 2: Class-specific uncertainty of BKL (left) and lesion class examples (right).

coincides with the observation of lesion class examples, in which AKIEC and DF are visually similar to BKL, while MEL and NV are different. It can be confirmed that the proposed class-specific uncertainty with TTMA provides information on between-class confusion.

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
