# OpenReview forum: "Test-Time Mixup Augmentation for Uncertainty Estimation in Skin Lesion Diagnosis"
_MIDL.io/2021/Conference/Short — MIDL 2021 Poster_

### Official Review · Reviewer_BY4t · 2021-04-20

**Confidence:** 4
**Final Rating:** 4

**Summary:**

The paper proposes a mixup-based uncertainty measure for the test-test augmentation approach. The mixup is conditioned on classes and is able to obtain both data uncertainty and class-specific uncertainty. The proposed approach is evaluated on the ISIC 2018 skin lesion diagnosis dataset and results show that the proposed TTMA improves accuracy, ECE, and out-of-distribution uncertainty.

**Strengths:**

1. The idea of class-conditioned mixup for uncertainty estimation is interesting and novel.
2. Empirical results show improved uncertainty on the ISIC 2018 skin lesion dataset as well as better out-of-distribution uncertainties.

**Weaknesses:**

My main concern is the lack of clear distinction between epistemic and aleatoric uncertainties: The referred approach TTA (Wang et al., 2019) is mainly proposed for aleatoric uncertainties, but this manuscript seems to argue for improved epistemic uncertainty. It is unclear whether the proposed approach uses MC-dropout for epistemic uncertainties, but I do not believe the proposed mixup approach provides direct estimates for the epistemic uncertainty, mainly due to the lack of a prior model distribution.

**Deanonymize Review:**

no

**Detailed Comments:**

1. The authors should clarify which uncertainty they are trying to improve and how exactly it is possible to improve the epistemic/model uncertainty without a model distribution.
2. For the performance comparison in Fig. 1 (1), it will be more informative to include the baseline accuracy and ECE without any test-test uncertainty enhancements.
3. TTA uses MC-dropout for the epistemic uncertainty; does the proposed approach uses MC-dropout, too?
4. In the first paragraph of section 2, the authors mention "we apply mixup augmentation on test data to obtain the perturbation-robust results." I don't understand how applying test-test mixup could result in more robust results as the mixup is only for obtaining uncertainties while the model parameters are unchanged. I'm wondering is mixup also used for training data or some adaptation/self-training is used for this claim?

**Justification Of The Rating:**

Although the manuscript lacks some important details regarding epistemic and aleatoric uncertainties, I think the proposed class-conditioned mixup uncertainty estimation is novel and of interest to the community.

**Paper Type:**

methodological development

**Special Issue:**

no

---

### Official Review · Reviewer_Aeix · 2021-04-25

**Confidence:** 5
**Final Rating:** 3

**Summary:**

This work proposes a novel model for uncertainty estimation of segmentation models. The authors extend the existing test-time augmentation (TTA) with using mixup for the augmentation. They showed that the proposed method can capture uncertainty information that is important for out-of-distribution detection. Experimental results with public dataset ISIC showed the effectiveness of this method. Overall this paper is novel and easy to understand. But some details are missing.

**Strengths:**

1. The idea of using mixup for test time augmentation is novel.
2. The authors applied uncertainty estimation for out-of-distribution detection, which is also a novel application.
3. The effectiveness of the proposed method was well validated with public datasets.


**Weaknesses:**

1. Some details of the implementation were not clear. For example, how many other images were used for mixup for one given test image? Was alpha a random value?
2. The authors only compared TTA with TTMA. The widely used uncertainty estimation Monte Caro Dropout was not compared in this paper.
3. The performance of TTA was much lower than TTMA in Fig. 1(a). This may depend on the authors’ implementation. What augmentation method was used for the existing TTA in this paper?
4. The authors did not provide visualization of uncertainty estimation, which makes it hard to know how the uncertainty estimation looks like.


**Deanonymize Review:**

yes

**Detailed Comments:**

Please see the above comments. The authors should clarify some important details, such as the implementation of TTMA and TTA. Reporting the performance of MC dropout is also expected, as it is an easy and widely used method for uncertainty estimation. A visualization of uncertainty would be very helpful for understanding.

**Justification Of The Rating:**

The idea of this paper is interesting. Both of the method and the applications are of high interest. The experimental results are basically convincing, but adding more details and some additional results would make this paper better.

**Paper Type:**

methodological development

**Special Issue:**

yes

---

### Meta-Review · Program_Chairs · 2021-05-06

**Recommendation:** Accept (Poster)
**Confidence:** 5

**Metareview:**

This paper is a clear acceptance. Authors are suggested to address reviewer suggestions in final version.

---

### Decision · Program_Chairs · 2021-05-11

Accept (Poster)